# Synaptic plasticity and cognitive function are disrupted in the absence of Lrp4

Andrea M Gomez, Robert C Froemke, Steven J Burden*

Graduate Program in Developmental Genetics, Molecular Neurobiology Program, Skirball Institute of Biomolecular Medicine, NYU Medical Center, New York, United States

**Abstract** Lrp4, the muscle receptor for neuronal Agrin, is expressed in the hippocampus and areas involved in cognition. The function of Lrp4 in the brain, however, is unknown, as $Lrp4^{-/-}$ mice fail to form neuromuscular synapses and die at birth. $Lrp4^{-/-}$ mice, rescued for Lrp4 expression selectively in muscle, survive into adulthood and showed profound deficits in cognitive tasks that assess learning and memory. To learn whether synapses form and function aberrantly, we used electrophysiological and anatomical methods to study hippocampal CA3–CA1 synapses. In the absence of Lrp4, the organization of the hippocampus appeared normal, but the frequency of spontaneous release events and spine density on primary apical dendrites were reduced. CA3 input was unable to adequately depolarize CA1 neurons to induce long-term potentiation. Our studies demonstrate a role for Lrp4 in hippocampal function and suggest that patients with mutations in $Lrp4$ or auto-antibodies to Lrp4 should be evaluated for neurological deficits.

## Introduction

In humans, mutations in genes encoding synaptic organizing complexes have been implicated in numerous and diverse neurological diseases, ranging from congenital myasthenia to autism spectrum disorders (*Sudhof, 2008*; *Burden et al., 2013*). Lrp4 plays a key role in the formation and maintenance of neuromuscular synapses, as a loss of Lrp4 leads to a failure to form neuromuscular synapses, and mutations in *Lrp4* or auto-antibodies to Lrp4 cause congenital myasthenia and myasthenia gravis, respectively (*Shen et al., 2013*; *Ohkawara et al., 2014*; *Tsivgoulis et al., 2014*). Lrp4 functions bidirectionally at neuromuscular synapses, where it responds to neuronal Agrin, stimulating MuSK, a receptor tyrosine kinase that functions as a master regulator of synapse formation, and functions in a retrograde manner to stimulate differentiation of motor nerve terminals (*Yumoto et al., 2012*).

Lrp4 belongs to the low-density lipoprotein receptor (LDLR) family, an ancient group of endocytic type 1, single-pass transmembrane proteins. Although LDLR family members were initially studied for their roles in receptor-mediated endocytosis, multiple other physiological roles have been described. Lrp4 has multifunctional roles in tissues other than the nervous system, including bone homeostasis, limb patterning, kidney formation, and placode development (*Johnson et al., 2005*; *Weatherbee et al., 2006*; *Ohazama et al., 2008*; *Li et al., 2010*; *Ahn et al., 2013*).

Lrp4 is expressed in the central nervous system (CNS) as well as in the peripheral nervous system (*Visel et al., 2004*; *Tian et al., 2006*; *Weatherbee et al., 2006*; *Lein et al., 2007*). Within the CNS, Lrp4 is expressed prominently in the hippocampus, olfactory bulb, cerebellum, and neocortex and present in postsynaptic membranes (*Tian et al., 2006*). The role of Lrp4 in the CNS is not understood, as *Lrp4* mutant mice die at birth from neuromuscular and respiratory failure, before synapse formation in the CNS ensues (*De Felipe et al., 1997*; *Tian et al., 2006*; *Weatherbee et al., 2006*; *Kim et al., 2008*; *Yumoto et al., 2012*). Previously, we generated mice that lack Lrp4 in all tissues except skeletal muscle and found that muscle-selective expression of Lrp4 ($Lrp4^m$) rescued the neuromuscular deficits

*For correspondence: steve.burden@med.nyu.edu

Competing interests: The authors declare that no competing interests exist.

**eLife digest** LRP4 is a muscle protein that is found in the hippocampus, a region of the brain that controls cognitive processes such as learning and memory. However, we know very little about what exactly LRP4 does in the hippocampus, and how it affects learning and memory.

A standard way to figure out what a protein does is to study mice that have been genetically modified so that they cannot produce that protein. However, deleting the gene for LRP4 leads to muscle problems that kill these mutant mice at birth.

To get around this problem, Gomez et al. have developed a method to restore the production of LRP4 in the muscles of mutant mice but not in their brains. These mutant mice were then subjected to a battery of tests to measure their ability to learn and recall new memories. These tests showed that LRP4 must be present in the brain, otherwise learning and memory are impaired.

Gomez et al. also explored a process known as long-term potentiation. This process, which involves strengthening the functional connections between neurons, is believed to be essential for learning and other cognitive process. Gomez et al. demonstrated that long-term potentiation was disrupted by the lack of LRP4.

Further experiments are needed to work out how LRP4 controls the learning process in the hippocampus and to explore the connection between LRP4 and various neuromuscular and neurological diseases.

of *Lrp4* mutant mice, allowing the mice to survive as adults (***Gomez and Burden, 2011***). To learn whether Lrp4 plays a role in the CNS, we used multiple behavioral paradigms to study the behavior of these muscle-rescued mice. Next, we examined the synaptic transmission and the anatomical organization of inputs onto CA1 hippocampal pyramidal neurons. Our data show that the rescued mice perform poorly in several learning and memory paradigms, demonstrating that Lrp4 has a critical role in the CNS. Moreover, we show that Lrp4 is enriched in postsynaptic membranes from the hippocampus, and our electrophysiological studies demonstrate a dramatic loss in long-term potentiation (LTP), accompanied by a reduction in synapses on apical dendrites of CA1 neurons.

## Results

### Lrp4 is required for associative learning and spatial memory

Newborn mice, which lack Lrp4 in all tissues except skeletal muscle ($Lrp4^{-/-}; Lrp4^m$), retained the fused digit and appendage defects found in *Lrp4* mutant mice. In other respects, the rescued mice appeared indistinguishable from their wild-type littermates (***Figure 1A***, inset). By three weeks after birth the growth rate of $Lrp4^{-/-}; Lrp4^m$ mice began to slow and by 6 weeks the mice were modestly runted (***Figure 1B***). Nonetheless, $Lrp4^{-/-}; Lrp4^m$ mice were fertile and lived a normal lifespan, indicating that Lrp4 is not required in tissues other than muscle for postnatal survival. The macroscopic morphology of the brain from adult $Lrp4^{-/-}; Lrp4^m$ mice appeared normal, although brain size, like body mass, was modestly reduced (***Figure 1C,D***).

Motor function is required to execute behavioral paradigms, thus we first asked if locomotion was normal in $Lrp4^{-/-}; Lrp4^m$ mice using an open-field test. When placed in the open-field arena, $Lrp4^{-/-}; Lrp4^m$ mice traveled as far and fast as control animals (***Figure 1E–G***), indicating that $Lrp4^{-/-}; Lrp4^m$ mice do not have obvious motor or skeletal defects. However, $Lrp4^{-/-}; Lrp4^m$ mice exhibited different open-field behavior compared to control animals (***Figure 1H,I***). The heat map in ***Figure 1I*** shows that control mice roamed throughout the open-field, whereas $Lrp4^{-/-}; Lrp4^m$ mice avoided the center of the arena (***Figure 1H,I***). Additionally, when suspended by the tail, $Lrp4^{-/-}; Lrp4^m$ mice display a stereotyped limb clasping behavior, similar to other mouse models of neurological disorders, including Rett Syndrome (***Figure 2***) (***Guy et al., 2001***).

To examine cognitive function and associative learning in these animals, we next assessed their behavior with classical fear-conditioning and passive avoidance paradigms (***LeDoux, 2003***; ***Lai and Ip, 2013***). In the fear-conditioning assay, mice were trained to associate an auditory cue with a foot shock, which elicited a freezing response. During training, both control and $Lrp4^{-/-}; Lrp4^m$ mice responded similarly to the tone and foot shock (***Figure 3A***), demonstrating that these sensory systems remained

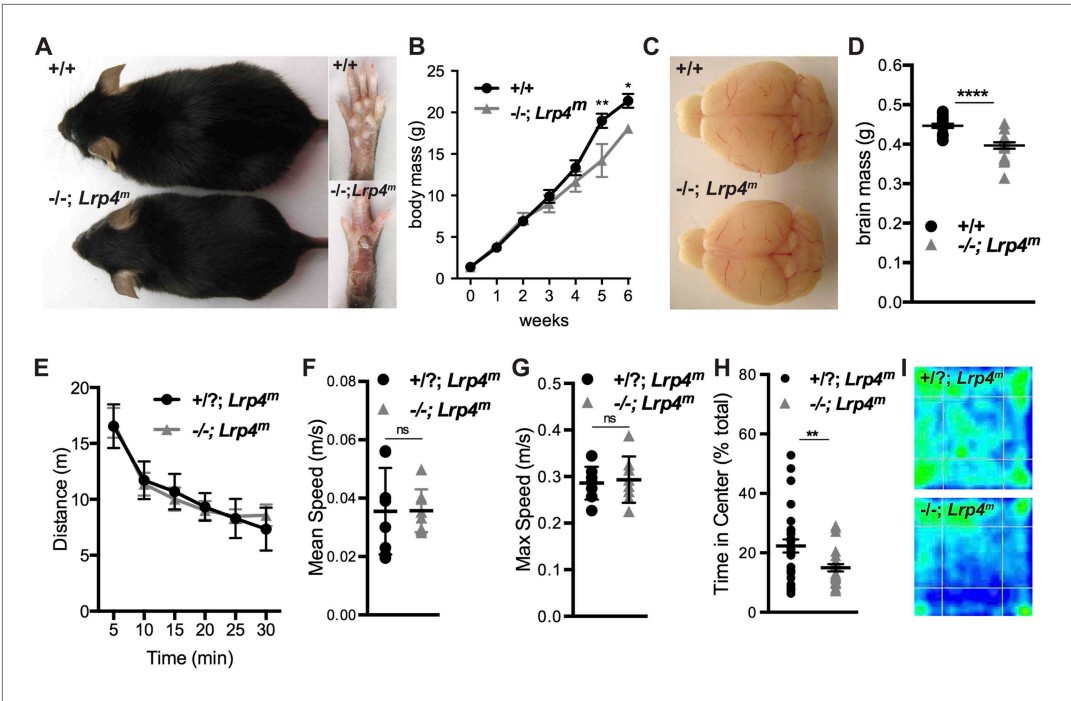

**Figure 1**. Restoring Lrp4 expression selectively in muscle of *Lrp4* mutant mice rescues neonatal lethality.
(**A**) *Lrp4*⁻/⁻; *Lrp4*ᵐ mice are fertile and live a normal lifespan. (**B**) The body mass of six-week old *Lrp4*⁻/⁻; *Lrp4*ᵐ mice is reduced by 16% (wild-type, 21.4 ± 0.8 g, n = 10; *Lrp4*⁻/⁻; *Lrp4*ᵐ, 18.0 ± 0.5 g, n = 5). (**C**) The gross morphology of the adult brain is similar in wild-type and *Lrp4*⁻/⁻; *Lrp4*ᵐ mice. (**D**) The size of the adult brain is reduced by 11% in *Lrp4*⁻/⁻; *Lrp4*ᵐ mice (wild-type, 0.45 ± 0.005 g, n = 17; *Lrp4*⁻/⁻; *Lrp4*ᵐ, 0.4 ± 0.008 g, n = 15). (**E**, **F**, **G**) The locomotor activity of *Lrp4*⁻/⁻; *Lrp4*ᵐ mice in an open field test was normal as measured by distance traveled (**E**), mean velocity (**F**), and maximum velocity (**G**) (wild-type, n = 17; *Lrp4*⁻/⁻; *Lrp4*ᵐ, n = 15). (**H**) *Lrp4*⁻/⁻; *Lrp4*ᵐ mice showed reduced exploratory behavior (wild-type, 22.3 ± 2.2%, n = 29; *Lrp4*⁻/⁻; *Lrp4*ᵐ, 14.9 ± 1.2%, n = 25). (**I**) Representative heat maps of wild-type and *Lrp4*⁻/⁻; *Lrp4*ᵐ mice during a 30 min open field test.

intact in the rescued mice. The next day, mice were exposed to the same tone in the absence of the foot shock, and the time spent freezing was measured. As expected, wild-type mice froze in response to the tone alone. In contrast, *Lrp4*⁻/⁻; *Lrp4*ᵐ mice spent much less time freezing, suggesting impaired learning or memory for the tone–shock pairing (***Figure 3A***).

We next assessed the rescued mice using a passive avoidance paradigm, which exploits an innate preference of mice to avoid a well-lit environment. Mice were placed in a well-lit chamber and freely allowed to enter a dark chamber, where they received a foot shock (***Figure 3B***). During training, control and *Lrp4*⁻/⁻; *Lrp4*ᵐ mice showed a similar latency to enter the dark chamber. 2 days later, mice were once again placed in a well-lit chamber, and the time to enter the dark chamber was recorded. Control mice were slow to enter the dark chamber after training, whereas *Lrp4*⁻/⁻; *Lrp4*ᵐ mice entered the dark chamber with a shorter latency than control mice, indicating an impaired association of the dark chamber with the foot shock (***Figure 3B***).

Fear conditioning and passive avoidance paradigms involve activity in the hippocampus among other brain areas (***LeDoux, 2003***). Because the hippocampus plays a major role in spatial learning,

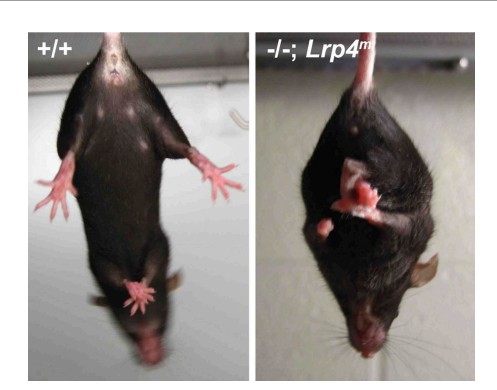

**Figure 2**. Forelimb and hindlimb clasping in *Lrp4*⁻/⁻; *Lrp4*ᵐ mice. Wild-type mice splay their limbs when suspended by their tail, whereas *Lrp4*⁻/⁻; *Lrp4*ᵐ mice clasp their limbs.

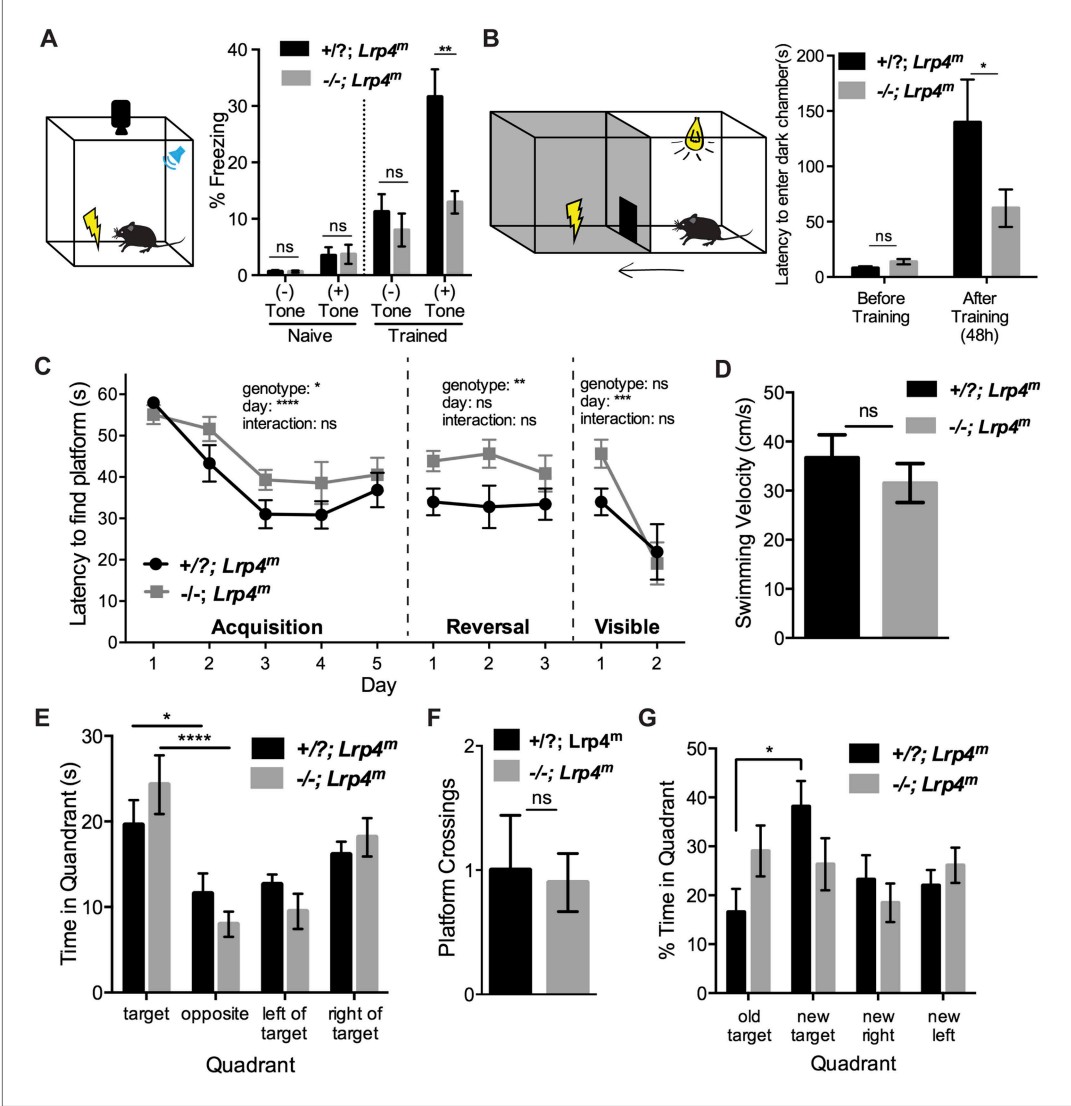

**Figure 3**. Mice lacking Lrp4 in the CNS display defects in learning and memory. (**A**) A schematic representation of the fear-conditioning paradigm. $Lrp4^{-/-}$; $Lrp4^m$ mice exhibit a decrease in freezing behavior, compared to littermate control mice, when presented with an aversive conditioned stimulus ($Lrp4^{-/-}$; $Lrp4^m$, n = 12; littermate controls, n = 14). (**B**) A schematic representation of the passive avoidance paradigm. $Lrp4^{-/-}$; $Lrp4^m$ mice were less hesitant to enter a dark chamber associated with an aversive stimulus ($Lrp4^{-/-}$; $Lrp4^m$, n = 13; littermate controls, n = 17). (**C**) $Lrp4^{-/-}$; $Lrp4^m$ mice showed spatial learning deficits and reduced cognitive flexibility in the Morris water maze. Both control and $Lrp4^{-/-}$; $Lrp4^m$ mice were able to locate the escape platform during the visible version of the water maze ($Lrp4^{-/-}$; $Lrp4^m$, n = 8; littermate controls, n = 9). (**D**) $Lrp4^{-/-}$; $Lrp4^m$ and control mice displayed comparable swimming velocity during the Morris water maze. (**E**) $Lrp4^{-/-}$; $Lrp4^m$ and control mice spent more time searching in the target quadrant region than other quadrants during the probe trial. (**F**) $Lrp4^{-/-}$; $Lrp4^m$ and control mice crossed the platform site with similar frequency during the probe trial. (**G**) $Lrp4^{-/-}$; $Lrp4^m$ mice spent less time in the new target quadrant during reversal training. See also **Figure 4**.

we used the Morris water maze to more specifically test whether $Lrp4^{-/-}$; $Lrp4^m$ mice have spatial learning and memory deficits. $Lrp4^{-/-}$; $Lrp4^m$ mice were trained for several days to associate visible spatial cues with the position of a hidden platform in an opaque pool (**Figure 3C** and **Figure 4A**). During this acquisition phase, the latency of control animals to locate the hidden platform decreased (**Figures 3C and 4B**). Although the latency for $Lrp4^{-/-}$; $Lrp4^m$ mice also decreased during this training period, they consistently took longer to find the platform than control mice (**Figures 3C and 4B**). After training, the hidden platform was removed and a trial was conducted to assess recall for the position of the hidden platform. Recall was

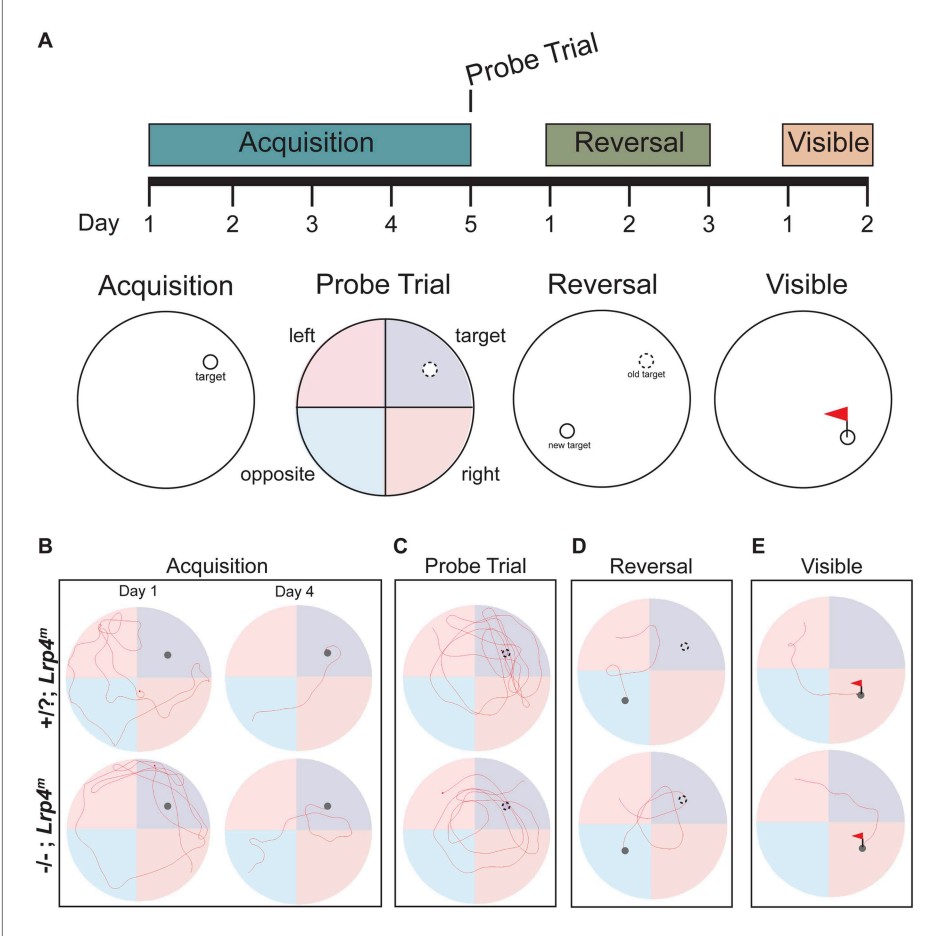

**Figure 4**. Mice lacking Lrp4 in the CNS display defects in learning and memory in the Morris water maze. (**A**) A schematic representation of the Morris water maze training protocol. Mice were trained for 5 days to locate a hidden platform. A probe trial was performed on the fifth day, when the platform was removed. The hidden platform was moved to the opposite quadrant during reversal training. A flag was placed on the hidden platform during the visible training phase. Representative trajectories of control and *Lrp4⁻/⁻; Lrp4ᵐ* mice during the acquisition (**B**), probe (**C**), reversal (**D**), and visible (**E**) tests of the Morris water maze trial.

quantitated by measuring the time spent in the target quadrant that previously contained the hidden platform. *Lrp4⁻/⁻; Lrp4ᵐ* mice displayed a preference for the correct target quadrant and crossed the former platform location with similar frequency to control mice (*Figures 3D and 4C*). These findings indicate that while the rescued mice had difficulty learning the position of the hidden platform; however, with repeated training, they were able to learn the position and eventually behave in a manner similar to control mice.

In order to assess cognitive flexibility of *Lrp4⁻/⁻; Lrp4ᵐ* mice, we placed a hidden platform in the quadrant opposite to the original target quadrant and tested the mice. Control mice rapidly learned the new location of the platform, whereas *Lrp4⁻/⁻; Lrp4ᵐ* mice showed a significant impairment (*Figures 3C and 4D*) and spent less time than control mice in the new target region (*Figure 3G*). These data indicate that the rescued mice were slower to extinguish their memory of the original target, learn the position of the new target, or both. Control and *Lrp4⁻/⁻; Lrp4ᵐ* mice were similarly able to locate a visible platform and had comparable swimming velocities, indicating that the impaired performance of *Lrp4⁻/⁻; Lrp4ᵐ* mice was not due to compromised visual acuity, swimming ability, or motivation to escape the water (*Figure 3C,D, 4E*).

## Hippocampal LTP is impaired without Lrp4

To determine whether the cognitive deficits observed in *Lrp4⁻/⁻; Lrp4ᵐ* mice are associated with synaptic dysfunction, we examined synaptic transmission using whole-cell recordings in acute hippocampal

slices. We focused on examining excitatory postsynaptic currents (EPSCs) in CA1 pyramidal cells evoked by stimulation of CA3 Schaffer collaterals (SC; *Figure 5A*) because of the extensive characterization of CA3–CA1 synapses in many mouse models (*Malenka and Bear, 2004*). We first determined whether Lrp4 is critical for synaptic transmission. SCs were briefly stimulated twice with varied interstimulus intervals, and EPSCs were recorded. At wild-type synapses, the second stimulus, delivered 50 ms later, elicited a greater postsynaptic response (*Figure 5B*, inset), possibly due to higher basal levels of residual calcium in nerve terminals following the first stimulus (*Katz and Miledi, 1968*; *Zucker and Regehr, 2002*). Paired-pulse facilitation was similar in control and *Lrp4$^{-/-}$; Lrp4$^m$* mice, indicating that Lrp4 is not essential for this form of evoked transmitter release and short-term plasticity (*Figure 5B*).

Next, we examined basal activity by measuring spontaneous miniature excitatory postsynaptic currents (mEPSC) in CA1 neurons. mEPSC frequency was reduced twofold in *Lrp4* mutant neurons (*Figure 5C,D*), although the mEPSC amplitude was normal (*Figure 5C,E,F*). The lower mEPSC frequency could be due to a reduced probability of vesicle fusion or a reduction in the number of synapses on CA1 neurons.

To determine whether Lrp4 is essential for induction or expression of LTP at CA3–CA1 synapses, where long-term synaptic plasticity is known to require postsynaptic depolarization and activation of postsynaptic NMDA receptors (Malenka and Nicoll, 1999), theta burst stimulation (TBS) was used to strongly activate SCs. EPSCs were recorded before and after repetitive TBS. TBS induced robust LTP from wild-type CA1 neurons (*Figure 5G,L,N*), whereas TBS failed to induce LTP in *Lrp4$^{-/-}$* CA1 neurons (*Figure 5H,M,N*). During the induction procedure the amplitudes of TBS-evoked EPSCs were reduced in *Lrp4$^{-/-}$* CA1 neurons compared to wild-type (*Figure 5I*). This dramatic loss of LTP demonstrates that Lrp4 is critical for a form of synaptic plasticity that has been linked to learning and memory.

We hypothesized that the smaller EPSCs during TBS in *Lrp4$^{-/-}$* CA1 neurons prevented LTP induction in these cells. Induction of LTP in CA1 neurons requires that SC-released glutamate binds to AMPA receptors to depolarize postsynaptic neurons to a level sufficient to release the $Mg^{2+}$-block of NMDA receptors and drive $Ca^{2+}$ influx into postsynaptic compartments (*Nowak et al., 1984*; *Collingridge et al., 1988*). To determine whether CA3 Schaeffer collaterals are unable to adequately depolarize CA1 neurons during TBS, we paired repetitive TBS with direct postsynaptic depolarization to 0 mV during LTP induction. We found that this brief period of depolarization of CA1 neurons was sufficient to restore normal TBS-induced LTP (*Figure 5J,K,M,N*). Importantly, these data show that expression of LTP in *Lrp4$^{-/-}$* CA1 neurons is intact as long as CA1 neurons are adequately depolarized during TBS. Thus, CA3 SCs appear unable to depolarize CA1 neurons to a level required to recruit NMDA receptors. Consistent with the idea that AMPA and NMDA receptors are available to participate in LTP, AMPA, and NMDA receptors are expressed at normal levels in synaptosomes isolated from *Lrp4$^{-/-}$; Lrp4$^m$* hippocampus (*Figure 6C,D*).

However, we also considered the possibility that in the absence of Lrp4, inhibition on CA1 neurons may have been strengthened, and depolarization-induced suppression of inhibition (DSI) may have removed the enhanced inhibition, unmasking LTP (*Pitler and Alger, 1994*; *Wilson et al., 2001*). We therefore directly measured inhibition to determine whether it was strengthened in *Lrp4$^{-/-}$; Lrp4$^m$* mice (*Figure 7*). Contrary to this notion, we found that inhibition was modestly reduced at low stimulation intensities and unchanged at higher stimulation intensities in *Lrp4$^{-/-}$; Lrp4$^m$* mice (*Figure 7G*). Because excitation was reduced while inhibition remained largely unaffected (*Figures 5,7*), the E/I ratio at CA1 synapses was diminished (*Figure 7H*). Together, these data are inconsistent with the idea that enhanced inhibition is responsible for a failure to elicit LTP and instead favor the idea that a failure of presynaptic input to adequately depolarize CA1 neurons underlies the LTP deficit.

## Spine density is reduced in CA1 primary apical dendrites

To assess whether the defects in synaptic transmission and failure of LTP induction were due to disorganization of hippocampal synaptic circuitry, we stained hippocampal slices from adult *Lrp4$^{-/-}$; Lrp4$^m$* hippocampus with probes that label nuclei, nerve endings, and excitatory postsynaptic membranes. The distribution of DAPI-stained nuclei and NeuN, a neuronal transcription factor, were comparable in slices from wild-type and *Lrp4$^{-/-}$; Lrp4$^m$* mice (*Figure 8A*). Additionally, the distributions of the presynaptic marker Synapsin and the excitatory postsynaptic marker PSD95 were similar in sections from wild-type and rescued mice (*Figure 8A*). Together, these data indicate that the defects in synaptic transmission and plasticity in *Lrp4$^{-/-}$; Lrp4$^m$* mice were not accompanied by a gross morphological disorganization of the hippocampus.

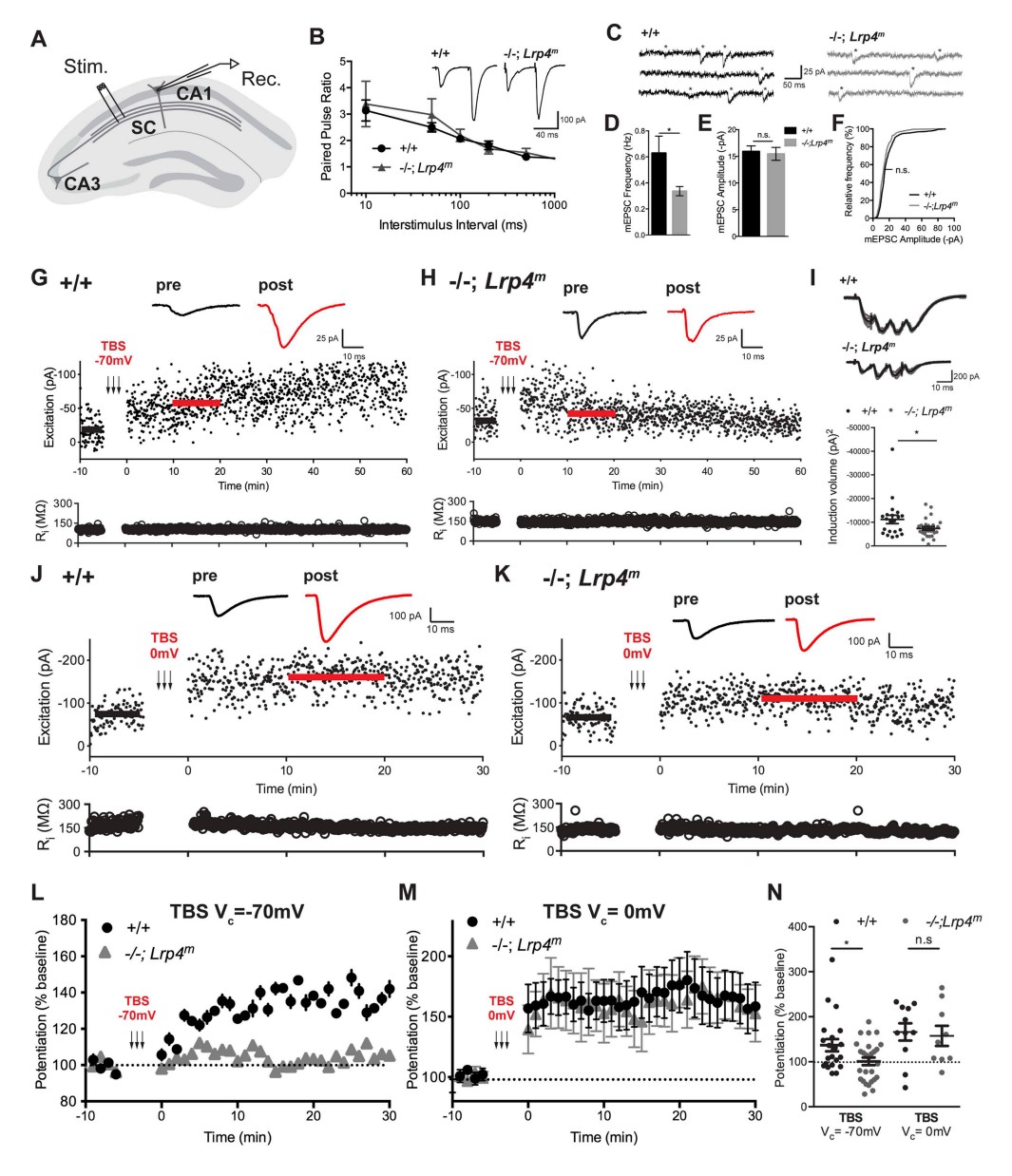

**Figure 5**. Lrp4 is required for normal synaptic transmission. (**A**) The panel shows the configuration of whole-cell voltage-clamp recordings made from acute hippocampal slices of young, adult mice. Postsynaptic responses in CA1 pyramidal neurons were measured following stimulation of Schaffer collaterals (SC). (**B**) Representative traces from CA1 neurons to paired stimuli show that paired-pulsed facilitation is normal in $Lrp4^{-/-}$; $Lrp4^m$ mice (wild-type, n = 9; $Lrp4^{-/-}$; $Lrp4^m$, n = 6). (**C**) Representative traces of spontaneous miniature excitatory postsynaptic currents (mEPSC) of wild-type or $Lrp4$ mutant CA1 neurons. (**D**) mEPSC frequency is reduced in $Lrp4$ mutant CA1 neurons (wild-type, n = 12; $Lrp4^{-/-}$; $Lrp4^m$, n = 12. (**E**, **F**) mEPSC ampitudes of $Lrp4$ mutant CA1 neurons are comparable to wild-type (wild-type, n = 12; $Lrp4^{-/-}$; $Lrp4^m$, n = 12). (**G**, **H**) Representative traces from individual CA1 neurons following a TBS delivered to neurons, which were voltage-clamped at −70 mV, show LTP from wild-type but not $Lrp4$ mutant neurons (upper panel). Excitation, measured as the amplitude of an EPSC (pA), is shown below, and input resistance (Ri) is shown in the bottom panel. There is considerable variability in the baseline EPSC amplitudes in slices from wild-type and $Lrp4^{-/-}$; $Lrp4^m$ mice. However, there was no significant difference in the baseline EPSC amplitudes between wild-type and mutant mice (wild-type, 52.6 ± 9.7 pA, n = 21; $Lrp4^{-/-}$; $Lrp4^m$, 44.9 ± 4.9 pA, n = 27, p = 0.44). Further, there was not a significant correlation between the magnitude of LTP and initial synaptic strength (wild-type, R2 = 0.05, n = 21, p = 0.35; $Lrp4^{-/-}$; $Lrp4^m$, R2 = 0.09, n = 21, p = 0.13). (**I**) Representative traces
*Figure 5. Continued on next page*

*Figure 5. Continued*

from individual CA1 neurons during TBS show a reduction in the integral of the summed EPCSs recorded during a single TBS in Lrp4-deficient neurons, top panel. Induction volume, quantified as the integral of the postsynaptic response, is shown in the bottom panel (wild-type, n = 21; *Lrp4*$^{-/-}$; *Lrp4*$^m$; n = 32) (**J, K**) Representative responses from individual CA1 neurons following a TBS delivered to neurons, which were voltage-clamped at 0 mV, show a restoration of LTP in *Lrp4* mutant neurons (upper panel). Excitation, measured as the amplitude of an EPSC (pA), is shown in the middle panel, and input resistance (Ri) is shown in the bottom panel. (**L**) Averaged data from control cells (n = 21) and *Lrp4* mutant (n = 32) neurons shows that the response from wild-type neurons is potentiated 1.5-fold, whereas *Lrp4* mutant CA1 neurons potentiate little, if at all 30 min following a TBS delivered at Vc = −70 mV. (**M**) Depolarization of *Lrp4* mutant CA1 neurons during TBS restores LTP (wild-type, n = 11; *Lrp4*$^{-/-}$; *Lrp4*$^m$, n = 9). (**N**) Quantitation of potentiation at 10-20 min after TBS demonstrates a lack of LTP in *Lrp4* mutant CA1 neurons, which is restored upon depolarization (Vc = −70 mV: wild-type, n = 21; *Lrp4*$^{-/-}$; *Lrp4*$^m$, n = 32; Vc = 0 mV: wild-type, n = 11; *Lrp4*$^{-/-}$; *Lrp4*$^m$, n = 9).

Dendritic spines contain the majority of excitatory synapses on CA1 pyramidal neurons, and changes in spine density lead to alterations in mEPSC frequency and are associated with aberrations in synaptic plasticity (*Sala et al., 2001*; *Tada and Sheng, 2006*). The inputs to CA1 pyramidal neurons are topographically organized at stereotyped positions along their apical–basal arborization (*Spruston, 2008*). Thus, we generated mice that are mutant for *Lrp4*, carry the *Lrp4*$^m$ transgene that restores Lrp4 expression in muscle, and also carry the *Thy1::YFP-H* transgene, which sparsely labels neurons, to examine spine density along the apical–basal axis (*Feng et al., 2000*). The spine density at several locations (basal, oblique apical, and tufts) was similar in *Lrp4*$^{-/-}$; *Lrp4*$^m$ and wild-type mice (*Figure 8B,C*). However, the density of spines on primary apical dendrites on CA1 neurons was reduced (~20%) in *Lrp4*$^{-/-}$; *Lrp4*$^m$ mice (*Figure 8B,C*). This decrease in spine density in CA1 neurons of *Lrp4*$^{-/-}$; *Lrp4*$^m$ mice is consistent with the reduction in mEPSC frequency and impaired synaptic plasticity.

## Lrp4 is enriched in synaptic membranes

During neuromuscular synapse formation, Lrp4 organizes synaptic differentiation by bidirectional signaling from the postsynaptic membrane (*Gomez and Burden, 2011*). To determine whether Lrp4 is enriched at synapses in the hippocampus we fractionated membranes from wild-type and *Lrp4* mutant hippocampus. We found that Lrp4 co-isolates with synaptosomes and is enriched in postsynaptic membranes, containing NMDA receptors, as well as presynaptic membranes, containing Synaptophysin (*Figure 6*).

Defects in LTP and changes in spine density are often associated with changes in the activity of NMDA receptors and/or AMPA receptors, two components critical for synaptic plasticity (*Zoghbi and Bear, 2012*; *Ebert and Greenberg, 2013*). To determine whether a loss of Lrp4 alters the expression of synaptic components, we measured the expression of presynaptic and postsynaptic proteins from adult hippocampal lysates (*Figure 6*). Wild-type and *Lrp4*$^{-/-}$ hippocampi expressed similar levels of NMDAR subunits, AMPAR subunits, synaptic scaffolding proteins, and cell-adhesion organizing molecules (*Figure 6*). These data indicate that the defects in synaptic transmission and cognition are not caused by a change in expression of key synaptic proteins.

## Discussion

Our study reveals that Lrp4 has a critical role in hippocampal function. Specifically, our results indicate that defects in synaptic transmission and postsynaptic integration may contribute to deficits in long-term plasticity, learning, and memory. Stimulation of CA3 inputs fails to induce LTP in *Lrp4*$^{-/-}$ CA1 neurons. Importantly, direct depolarization of *Lrp4*$^{-/-}$ CA1 neurons during TBS can rescue LTP. Thus, CA3–CA1 synapses in *Lrp4* mutant neurons have the capacity to express LTP. Because direct stimulation by-passes the normal synaptic mechanisms for depolarizing CA1 neurons, the absence of LTP in *Lrp4*$^{-/-}$ CA1 neurons may be due to a failure of SC inputs to adequately depolarize CA1 neurons and remove the Mg$^{++}$-dependent block of NMDA receptors. The reduction in the number of CA3–CA1 synapses on apical dendrites is consistent with this view.

We do not yet know when and where Lrp4 is required for normal CNS function. Further studies will be required to learn whether Lrp4 is required for neurogenesis, early steps in synapse formation, later stages in synaptic differentiation, and the construction and function of Lrp4-dependent circuits that

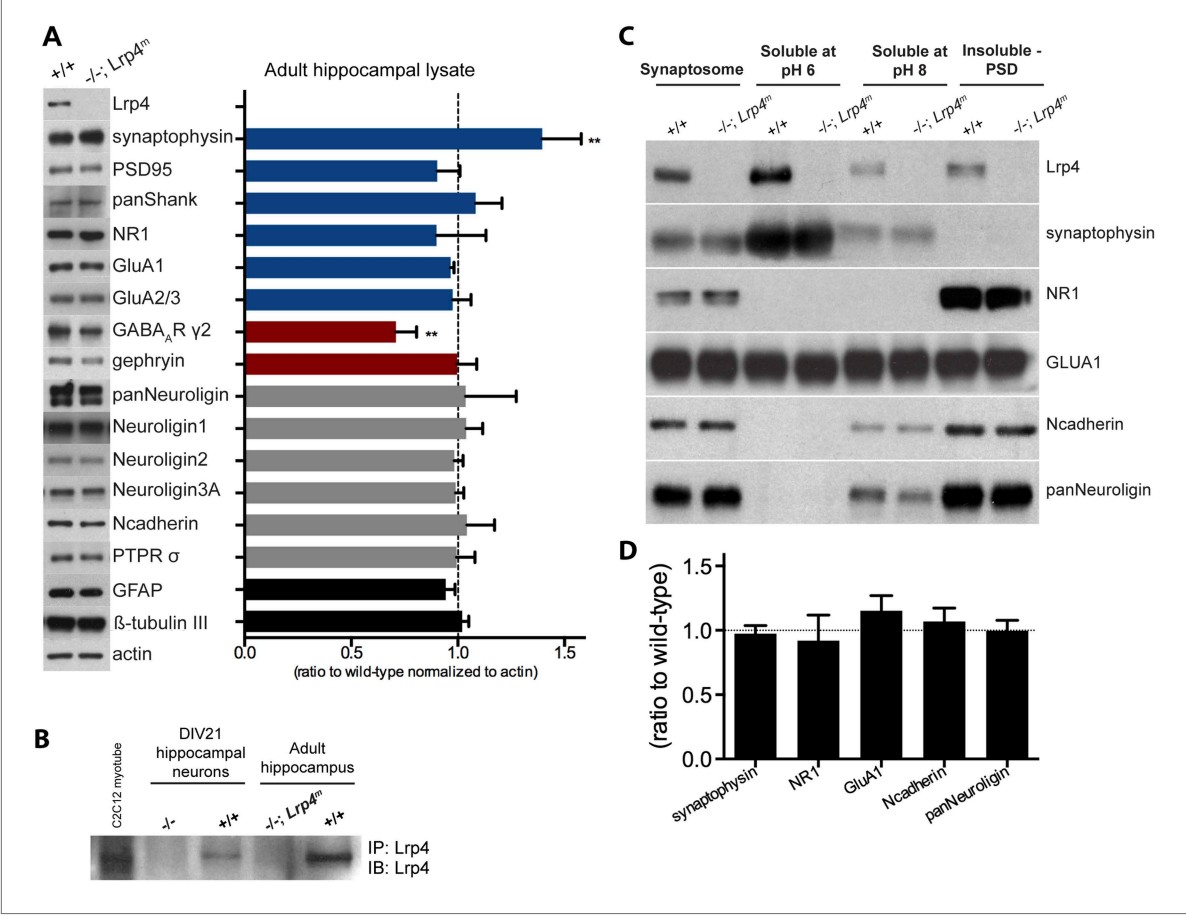

**Figure 6**. Lrp4 is enriched in synaptic membranes. (**A**) Quantitative analysis of proteins in hippocampal lysates from wild-type and *Lrp4⁻/⁻; Lrp4ᵐ* mice. The expression level of each protein, normalized to actin, was determined and assigned a value of 1.0 in wild-type mice. The graph shows the ratio of values in *Lrp4⁻/⁻; Lrp4ᵐ* mice compared to wild-type mice. Expression of most proteins is not dependent upon Lrp4, but expression of the synaptic vesicle-associated protein, Synaptophysin, was modestly elevated (30%) and expression of the GABAARγ2 subunit was modestly decreased (20%) in *Lrp4* mutant hippocampi. (**B**) Lrp4 expression is detected in cultured hippocampal neurons, grown in cell culture for 21 days, and in hippocampal tissue from wild-type but not from *Lrp4⁻/⁻; Lrp4ᵐ* mice. (**C**) Lrp4 co-isolates with synaptosomes, the pH 6-solubilzed fraction, which is enriched for Synaptophysin, a presynaptic marker, and the postsynaptic density (PSD) fraction, which is highly enriched for NR1. (**D**) The expression levels of presynaptic and postsynaptic proteins present in the synaptosomal fraction (**C**) are not altered in *Lrp4⁻/⁻; Lrp4ᵐ* mutant hippocampi.

are important for cognition. It seems likely that the loss of neuronal Lrp4 is responsible for the changes in cognition and synaptic function described here. Consistent with this idea, *Lrp4* is strongly expressed in the dentate gyrus and CA fields of the hippocampus, as well as in the olfactory bulb, and cerebral cortex (**Tian et al., 2006**; **Lein et al., 2007**). Moreover, Lrp4 co-fractionates with the postsynaptic density (**Tian et al., 2006**) (**Figure 6C**). Further, Lrp4 is expressed on the cell surface of cultured cortical neurons (**Tian et al., 2006**). Nonetheless, it may also be the case that other cell types, including glia, may express Lrp4 and contribute to the cognitive deficits found in muscle-rescued *Lrp4* mutant mice. Because glia are known to regulate the efficiency of synaptic transmission (**Eroglu and Barres, 2010**), a potential defect in glia–neuron signaling could contribute to a failure of synaptic transmission, post-synaptic integration, and/or synapse formation on CA1 apical dendrites.

*Lrp4*-deficient mice rescued for neuromuscular synapses appeared to have a range of cognitive defects. Mice displayed anxiety-like behavior in the open-field test and perseverative behavior in the Morris water maze. Stereotypic and restricted repetitive behaviors are symptoms often observed in patients with autism (**Lord et al., 2000**) and in mouse models of autism spectrum disorders (**Banerjee et al., 2014**). Accordingly, it will be interesting to explore whether *Lrp4*-deficient mice are predisposed to additional autism-like phenotypes such as altered social behavior, hyperactivity, and epilepsy.

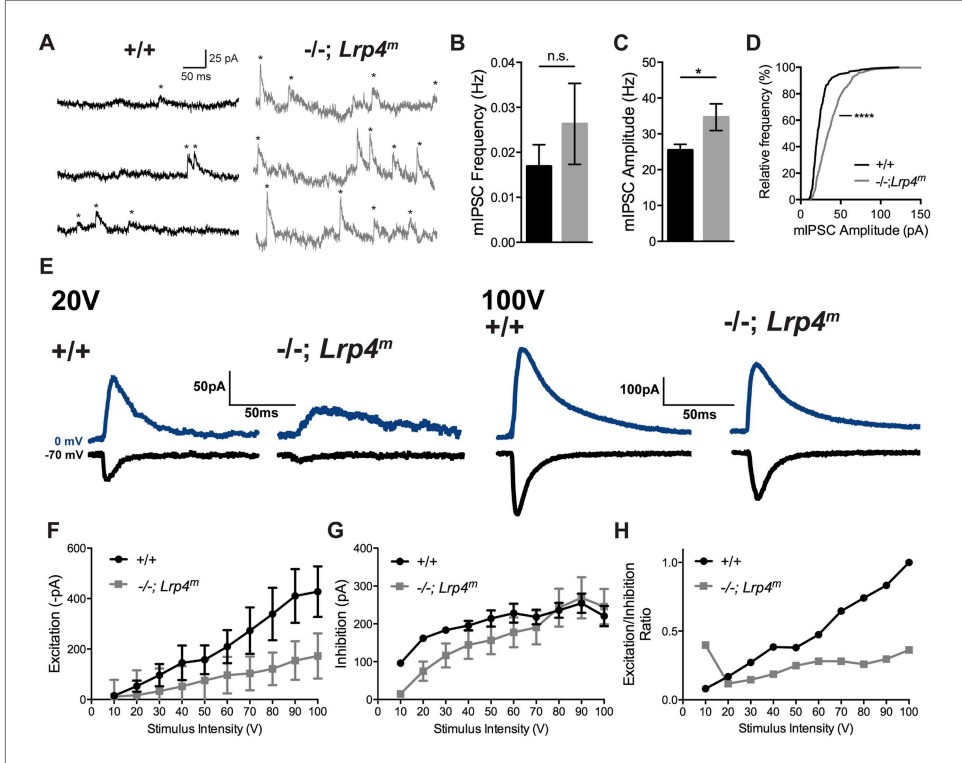

**Figure 7**. The strength of inhibition is unchanged in CA1 neurons from $Lrp4^{-/-}$; $Lrp4^m$ mice. (**A**) Representative traces of spontaneous miniature inhibitory postsynaptic currents (mIPSC) from CA1 neurons from wild-type or $Lrp4^{-/-}$; $Lrp4^m$ mice, which were voltage-clamped at 0 mV. (**B**) The mIPSC frequency is similar in Lrp4 mutant and wild-type CA1 neurons (wild-type, 0.017 ± 0.005 /sec, n = 7; $Lrp4^{-/-}$; $Lrp4^m$, 0.026 ± 0.009 /sec, n = 6). (**C, D**) mIPSC amplitudes are increased in CA1 neurons from $Lrp4^{-/-}$; $Lrp4^m$ mice (wild-type, 25.4 ± 1.6 pA, n = 7; $Lrp4^{-/-}$; $Lrp4^m$, 34.7 ± 3.7 pA, n = 6). (**E**) Representative traces of excitation (black) and inhibition (blue) at varied stimulus intensities. (**F**) Excitation is decreased in CA1 neurons from $Lrp4^{-/-}$; $Lrp4^m$ mice. (**G**) Inhibition is similar in CA1 neurons from wild-type and $Lrp4^{-/-}$; $Lrp4^m$ mice. (**H**) The E/I ratio is decreased in CA1 neurons from $Lrp4^{-/-}$; $Lrp4^m$ mice.

Mutations in *Lrp4* are responsible for Cenani–Lenz syndrome, characterized by bone abnormalities and fusions in hand, limb, and other bones (*Karner et al., 2010*; *Li et al., 2010*; *Kariminejad et al., 2013*; *Khan et al., 2013*). Lrp4 binds, sequesters, and presents negative regulators of Wnt- and BMP-signaling, such as Dickhopf, Sclerostin, and Wise (*Ohazama et al., 2008*; *Choi et al., 2009*; *Ahn et al., 2013*). In certain instances, Cenani–Lenz syndrome is caused by mutations that prevent Lrp4 from interacting with these negative regulators, leading to excessive Wnt and/or BMP signaling (*Leupin et al., 2011*). Mutations in *Lrp4* that reduce Agrin–Lrp4–MuSK signaling, without perturbing Wnt signaling, cause a neuromuscular disease, termed congenital myasthenia (*Ohkawara et al., 2014*). Moreover, auto-antibodies to Lrp4 are responsible for one form of myasthenia gravis (*Higuchi et al., 2011*; *Zisimopoulou et al., 2014*). It is currently unclear whether patients with Cenani–Lenz syndrome, *Lrp4* congenital myasthenia, or auto-immune Lrp4 myasthenia gravis have cognitive deficits or neurological complications. Given our data showing that Lrp4 has an important role in cognition in mice, it will be interesting and important to evaluate the neurological status of these patients.

## Materials and methods

### Mice
Mice that are null for *Lrp4*, or carry muscle-specific::*Lrp4* (*Lrp4^m*) or *Thy1::YFP-H* transgenes, have been described previously (*Feng et al., 2000*; *Weatherbee et al., 2006*; *Gomez and Burden, 2011*). Procedures were approved by the New York University School of Medicine Institutional Animal Care and Use Committee (Protocol 140406-01).

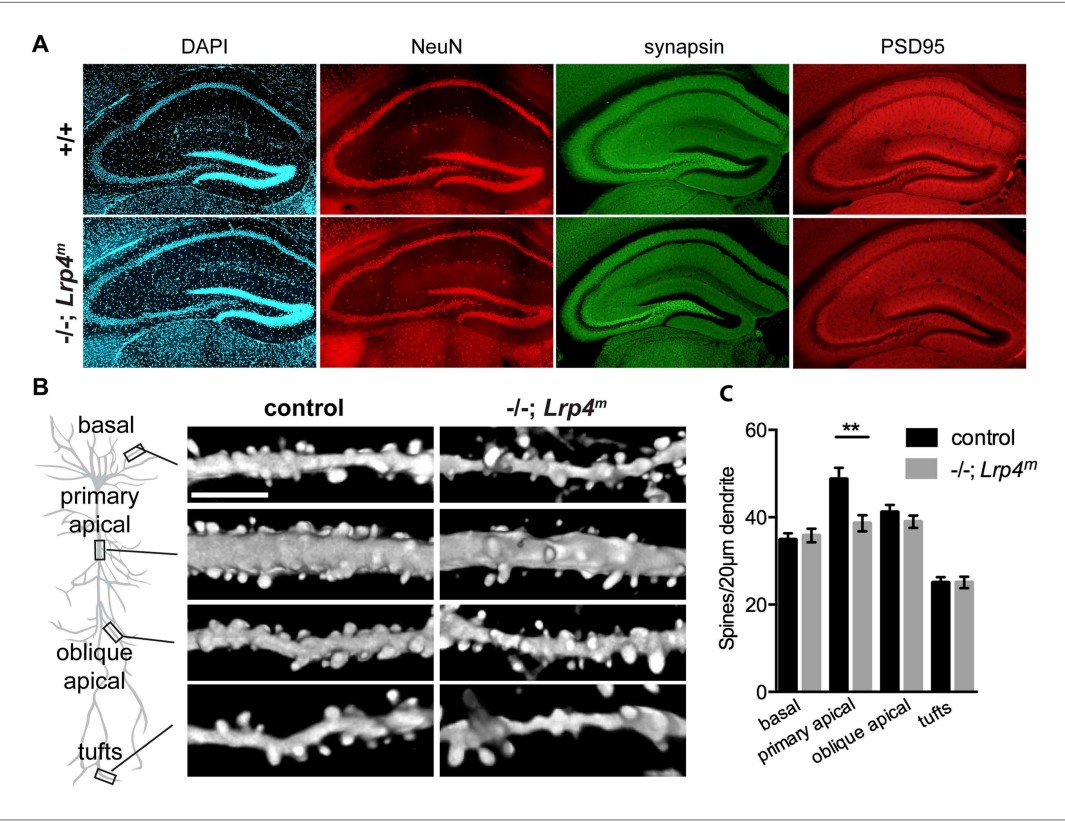

**Figure 8**. A loss of Lrp4 decreases spine density in primary apical dendrites of CA1 neurons. (**A**) The organization of the hippocampus appears normal as assessed by staining sections of the adult hippocampus for DNA (DAPI), neuronal nuclei (NeuN), presynaptic terminals (synapsin) or excitatory postsynaptic membranes (PSD95). (**B**) Representative images of dendrites of Thy1-YFP labeled CA1 Lrp4 mutant pyramidal neurons near (basal and apical) and far (oblique apical and tufts) from the cell body. Scale bar: 5 μm. (**C**) The spine density is reduced selectively at primary apical dendrites (basal: wild-type, n = 45; $Lrp4^{-/-}$; $Lrp4^m$, n = 45; primary apical: wild-type, n = 43; $Lrp4^{-/-}$; $Lrp4^m$, n = 46; oblique apical: wild-type, n = 42; $Lrp4^{-/-}$; $Lrp4^m$, n = 46; tufts: wild-type, n = 42; $Lrp4^{-/-}$; $Lrp4^m$, n = 45).

## Open field test

Mice were placed in an open field box (40 × 40 × 30 cm) for 30 min, and movement was recorded and analyzed with ANY-maze video tracking software (Stoelting, Wood Dale, IL). Open fields were thoroughly washed with water and ethanol between sessions.

## Fear conditioning test

Mice were trained and tested in a sound-attenuated cage using the FreezeFrame system (Coulbourn Instruments, Whitehall, PA), and behavior was recorded using low-light video cameras. Stimulus presentation was automated using Actimetrics FreezeFrame software (version 2.2; Coulbourn Instruments). Test cages were equipped with stainless-steel shocking grids, which were connected to a precision feedback current-regulated shocker (Coulbourn Instruments). Mice were allowed to roam for 2 min, without a shock, on a stainless-steel shocking grid connected to a precision feedback current-regulated shocker in ethanol-scented cages (Coulbourn Instruments). Fear conditioning was established by three auditory tone (30 s, 4000 Hz, 80 dB)/foot shock (2 s, 0.5 mA) pairings, separated by 15 s without stimuli. 2 min after conditioning, mice were returned to their home cages. 24 hr after training, the mice were placed in a 1% Pine-Sol-scented cage on non-shocking grids with a different texture than the stainless-steel shocking grids used during training. Following 2 min without an auditory stimulus, mice were presented with the training tone (4000 Hz, 80 dB) for 2 min. The percent of time spent in a frozen stature, before and after training, was measured. All equipments were thoroughly cleaned with detergent and water between sessions.

## Passive avoidance test

Mice were trained and tested in a two-chamber passive avoidance cage, in which one chamber was permanently darkened and separated from the lit chamber by a sliding door. The darkened chamber was equipped with a stainless-steel shocking grid connected to a precision feedback current-regulated shocker (Coulbourn Instruments). Equipment control was automated with Graphic State (Coulbourn Instruments). During training, mice were placed in the brightly-illuminated chamber and allowed to move freely for 1 min. Following this period, the door to the dark chamber was opened, and mice were allowed to move into the dark chamber. Latency to enter the dark chamber was recorded. Once the mice had completely entered the dark chamber, the door shut and a 2 s, 0.5 mA shock was delivered. After 10 s in the dark chamber, mice were returned to their home cage. During the test period, 48 hr after training, the training protocol was repeated, but the mice did not receive a foot shock upon entry into the dark chamber. Latency to enter the dark chamber was recorded. All equipment was thoroughly cleaned with detergent and water between sessions.

## Morris water maze

Mice were given four trials (60 s/trial; swim-start position randomized) each day to find a hidden platform in a circular pool of water, rendered opaque with white tempera paint, using visual cues placed outside the pool. The trajectories of mice were recorded and analyzed with Ethovision XT software (Noldus, Attleboro, MA). The time required to find the escape platform (escape latency) was measured. At the end of the fifth day, the platform was removed and a 1 min probe trial was conducted to measure the percent of time spent in each quadrant as well as the frequency which mice crossed the former location of the platform. For reversal training, the hidden platform was moved to the opposite quadrant, and mice were given four trials (60 s/trial; swim-start position randomized) for 3 days to locate the new platform position. Escape latencies from four trials each day were recorded. To control for motivation to escape the water, visual acuity, and swimming ability mice were trained on a visible platform for 2 days (60 s/trial; swim-start position randomized).

## Electrophysiology

Coronal hippocampal slices (350 μm) were prepared from one hemisphere of age-matched mice (P21–P34) anesthetized with intraperitoneal injection of ketamine/xylazine (ketamine, 100 mg/kg; xylazine, 10 mg/kg). Slices were cut with a vibratome (VT1200S, Leica, Buffalo Grove, IL) and placed in ice-cold oxygenated (95% $O_2$/5% $CO_2$) dissection buffer, which was (in mM): 75 sucrose, 87 NaCl, 2.5 KCl, 1.25 $NaH_2PO_4$, 0.5 $CaCl_2$, 7 $MgCl_2$, 25 $NaHCO_3$, 1.2 ascorbic acid, and 10 dextrose, pH 7.4. After approximately 30 min, the solution was gradually warmed to room temperature. Slices were then transferred to artificial cerebrospinal fluid (ACSF), which was (in mM): 124 NaCl, 2.5 KCl, 1.25 $NaH_2PO_4$, 2.5 $CaCl_2$, 2 $MgSO_4$, 26 $NaHCO_3$, 10 dextrose, and 4 sucrose, and incubated at room temperature for at least 30 min to allow for recovery. Slices were then transferred to the recording chamber and perfused (2.0–2.5 ml/min) with oxygenated ACSF at 32°C with a TC-344B in-line solution heater and controller (Warner Instruments, Hamden, CT). Somatic whole-cell recordings were made from CA1 pyramidal hippocampal neurons, which were voltage clamped with an Axoclamp 2B amplifier (Molecular Devices, Sunnyvale, CA) and imaged using infrared-differential interference contrast video microscopy, digitized by Digidata 1440a (Molecular Devices). Patch pipettes (4–8 MΩ) were filled with (in mM): 125 Cs-gluconate, 2 CsCl, 5 TEA, 4 ATP, 0.3 GTP, 10 phosphocreatine, 10 HEPES, 0.5 EGTA, and 3.5 QX-314. Data were filtered at 2 kHz, digitized at 10 kHz, and analyzed with Clampfit 10 (Molecular Devices). SC afferents were stimulated with a small glass bipolar electrode (S88 Stimulator, Grass Instruments, Warwick, RI). Paired-pulse facilitation was induced with two stimuli of equal intensity presented at variable interstimulus intervals, ranging from 10 ms to 1 s, and quantified as the ratio of second to first EPSC. Once a baseline for synaptic transmission was stable for 10 min, LTP was induced with a TBS. TBS consisted of four trains, separated by 20 s intervals. Each train was comprised of ten bursts at 5 Hz, and each burst included 4 stimuli at 100 Hz. During TBS, CA1 neurons were held either at the resting potential (−70 mV) or depolarized to 0 mV. We computed the magnitude of LTP as the average synaptic strength 10–20 min after pairing, normalized by the average synaptic strength before pairing (*Feldman, 2000*; *Froemke et al., 2005*; *Gambino et al., 2014*). During TBS, we integrated four consecutive EPSCs and termed this value the TBS-evoked EPSC; for each cell, all TBS-evoked EPSCs were averaged, and the mean value was determined (*Figure 3I*). We measured inhibitory

postsynaptic currents by voltage-clamping CA1 neurons to 0 mV, near the reversal potential for excitation.

## Statistics

Data are presented as mean ± SEM. p values are derived from unpaired, two-tailed Student's $t$-tests or two-way analysis of variance (ns = not significant, $*p < 0.05$, $**p < 0.01$, $***p < 0.001$, $****p < 0.0001$).

## Protein fractionation and Western blotting

Hippocampi were isolated in cold PBS, flash frozen in liquid nitrogen, and stored at −80°C. Brain lysates, synaptosomes, and PSD fractions were prepared as described previously (*Phillips et al., 2001*; *Jordan et al., 2004*). An equal amount of protein (10 µg) from each fraction was separated by SDS-PAGE. The following antibodies were diluted in TBST (0.2% Tween-20), with 2% BSA: Lrp4 (1:2500; Neuromab, N207/27, Davis, CA), Synaptophysin (1:20,000; Life Technologies, Grand Island, NY), PSD95 (1:1000; Neuromab, K28/43), GluA1 (1:1000; Millipore, Billerica, MA), GluA2/3 (1:500; Millipore), pan-Shank (1:1000; Santa Cruz Biotechnology, C-20, Dallas, TX), NR1 (1:1000; Neuromab, N308/48), $GABA_A R$ γ2 (1:1000; PhosphoSolutions, Aurora, CO), NeuN (1:1000; Millipore), Gephyrin (1:1000; SYnaptic Systems, Goettingen, Germany), actin (1:2000; Sigma, AC74, St. Louis, MO), N-cadherin (1:1000; BD Transduction Laboratories, 610920, San Jose, CA), PTPR σ (1:500; Protein Tech Group, Chicago, IL), GFAP (1:1000; Sigma), β-tubulin III (1:6000; SYnaptic SYstems). The following antibodies were gifts from P. Scheiffele: Neuroligin 1 (1:2000), Neuroligin 2 (1:2000), Neuroligin 3A (1:2000), and pan-Neuroligin (1:3000).

## Immunohistochemistry

Deeply anesthetized (ketamine, 100 mg/kg; xylazine, 10 mg/kg, i.p.) mice were transcardially perfused briefly with phosphate buffer saline (PBS) followed by 4% paraformaldehyde in PBS. Coronal slices (40 µm) were blocked and permeabilized in PBS with 2% normal goat serum and 0.2% Triton X-100, followed by overnight incubation at 4°C with primary antibodies. The sections were subsequently washed in PBS and incubated with secondary antibody.

## Hippocampal cell culture

Dissociated primary hippocampal neuron cultures were prepared from embryonic day 18 mouse embryos, as described previously (*Osten et al., 1998*). Neurons were plated at a density of $1 \times 10^6$ cells in poly-L-lysine coated 60-mm tissue culture dishes and grown in Neurobasal Medium, supplemented with B-27 (Life Technologies). 3 days after plating cells, 2 µM Ara-C was added to the medium to minimize growth of dividing cells. The medium, including Ara-C, was replaced once per week.

# Acknowledgements

We thank J Dasen and RW Tsien for helpful comments on the manuscript, W Gan, T Franke, C Hoeffer, L Perez-Cuesta, C Lai, and C Wincott for assistance with the behavioral studies and spine imaging, P Scheiffele for kindly providing antibodies, and RW Tsien for suggesting experiments. The authors declare no competing financial interests. AMG. is currently at the Biozentrum, Basel, Switzerland.

# Additional information

## Funding

| Funder | Grant reference number | Author |
|---|---|---|
| Alfred P. Sloan Foundation (Sloan Foundation) | Robert Froemke | Robert C Froemke |
| National Institute of Neurological Disorders and Stroke | NS36193 | Steven J Burden |
| National Cancer Institute | P30CA16087 | Andrea M Gomez, Robert C Froemke, Steven J Burden |

The funders had no role in study design, data collection and interpretation, or the decision to submit the work for publication.

## Author contributions

AMG, Conception and design, Acquisition of data, Analysis and interpretation of data, Drafting or revising the article; RCF, SJB, Conception and design, Analysis and interpretation of data, Drafting or revising the article

## Ethics

Animal experimentation: All procedures were approved by the New York University School of Medicine Institutional Animal Care and Use Committee (Protocol 140406-01).

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
