## [Decision Letter]

Thank you for submitting your work entitled “Synaptic Plasticity and Cognitive Function Are Disrupted in the Absence of Lrp4” for consideration by *eLife*. Your article has been favorably evaluated by a Senior editor, a Reviewing editor, and two reviewers. There are just a few issues that need to be addressed before acceptance. Related to the point 1 of reviewer 1, we do not expect you to provide data on a forebrain specific KO, unless you have already obtained data from this model.

*Reviewer*
*1:*

The study by Gomez et al. reports that knockout of Lrp4 in all tissues in mice except skeletal muscle (where it is required for survival), causes impairment in hippocampal-dependent learning and memory behavioral tasks, LTP at the Schaffer collateral, and a reduction in spine density in CA1 primary apical dendrites.

1) Although the behavioral tasks are carefully controlled for motor deficits, the Lrp4m mice have fusiform digits and appendage defects, and reduced body weight, which makes the reader wonder whether defects outside the hippocampus might account for some of the behavioral phenotypes. Did the authors consider a forebrain-specific knockout? I'm not suggesting this as a revision experiment, but it would be helpful if the authors comment on other effects knockout of Lrp4 might have (e.g. ability to swim in a straight line, vision, hearing, gait and muscle power). It would be even more convincing if one of the behavioral or electrophysiological (or spine number) effects could be rescued by hippocampal expression of Lrp4.

2) In Figure 1; time spent in the center quadrant in the open field test, it doesn't look like a significant difference between wild-type and Lrp4 mutant mice from the data scatter or error bars. Are these really significantly different?

3) In Figure 5 G, and H; wild-type mice have approximately -20 pA pre TBS amplitude, and ko approximately -30 pA. Although these can't be compared directly, it makes one wonder if there is a difference in evoked synaptic transmission/ input-output curves for fiber volley. This is important to test because the decrease in mEPSC frequency and spine density suggest a decrease in synapse number in the Lrp4m mutants.

4) Also in Figure 5, panel G and H, post TBS, the red bar in wt is approximately -55 (compared to -20 pre TBS), and in ko is -40 (compared to -30 pre TBS), but the % baseline values in L don't seem to reflect these values. Can the authors describe how they normalized panels G and H to generate panel L? Likewise, in panel J there's a change from -70 to -160 (black bar compared to red bar) in wt and in panel K from -70 to -120 in ko, which may be significantly different. But in the comparison in panel M, both overlap at approximately 160% potentiation?

5) Do the authors propose two different effects of Lrp4? In synapse formation and plasticity? Or do they think the mEPSC frequency decrease and spine number decrease are related to the LTP deficit? To test synapse formation in more detail they could quantify number of synapses (i.e. pre and post-synaptic sites marked with synaptophysin or PSD-95; in slices or in hippocampal cultures) in wild-type and Lrp4m. This would also distinguish between the mEPSC frequency being caused by a change in synapse number or release probability.

6) A possible cause of the LTP deficit in Lrp4m kos, on the other hand, could be a decrease in the strength of excitatory synapses in knockouts. In the Discussion the authors state that inhibition may have been strengthened in Lrp4 kos. What is the basis for this statement? There was a significant decrease in GABA in the kos (which is not commented on). It would be useful to measure inhibition, and possible changes in GABA at synapses in addition to test for possible changes in excitation/ inhibition ratio, which could explain the LTP phenotype. In addition, it would be helpful to know if Lrp4 is expressed in all neurons or in particular subsets of neurons in the hippocampus where it might change feedback or feed-forward inhibition, for example, or if it acts more generally at all synapses.

*Reviewer*
*2:*

This is a very well designed study documenting importance of LRP4 in central nervous system and especially for long-term plasticity, learning and memory. The findings are based on well-designed behavioral function tests, direct recordings from hippocampal neurons, and examination of dendritic ultrastructure. The altered synaptic plasticity is attributed to decreased dendritic spine density and reduced mEPSC frequency. I only have two comments, one of which is minor.

1) Avoid abbreviations in the Abstract (e.g., spell out LTP)

2) LRP4-seropositive myasthenic patients are unlikely to develop neurological deficits in central nervous system because autoantibodies do not cross the blood-brain barrier unless it is inflamed or otherwise damaged. This interpretation should be omitted in the Abstract and the Discussion.

---

## [Author Response]

Reviewer 1:

*The study by Gomez et al. reports that knockout of Lrp4 in all tissues in mice except skeletal muscle (where it is required for survival), causes impairment in hippocampal-dependent learning and memory behavioral tasks, LTP at the Schaffer collateral, and a reduction in spine density in CA1 primary apical dendrites*.

*1) Although the behavioral tasks are carefully controlled for motor deficits, the Lrp4m mice have fusiform digits and appendage defects, and reduced body weight, which makes the reader wonder whether defects outside the hippocampus might account for some of the behavioral phenotypes. Did the authors consider a forebrain-specific knockout? I'm not suggesting this as a revision experiment, but it would be helpful if the authors comment on other effects knockout of Lrp4 might have (e.g. ability to swim in a straight line, vision, hearing, gait and muscle power). It would be even more convincing if one of the behavioral or electrophysiological (or spine number) effects could be rescued by hippocampal expression of Lrp4*.

Due to the bone abnormalities of *Lrp4*^*-/-*^*; Lrp4*^*m*^ mice, their gait was visibly distinct from control mice. Nonetheless, in the open-field test, *Lrp4*^*-/-*^*; Lrp4*^*m*^ mice performed as well as control mice in distance traveled and maximum speed (Figure 1). Moreover, in the Morris water maze, *Lrp4*^*-/-*^*; Lrp4*^*m*^ mice swam directly toward the visible platform at a speed that was comparable to control mice (Figure 3), indicating that neither vision nor muscle power were noticeably impaired. Together, these data support the idea that their defects in learning in behavior are unlikely due to a motor deficit.

Although we did not test directly for audition, *Lrp4*^*-/-*^*; Lrp4*^*m*^ mice learned to associate a tone with a foot shock, albeit less well than control mice (Figure 1). These findings indicate that *Lrp4*^*-/-*^*; Lrp4*^*m*^ mice sensed the auditory tone.

*2) In*
Figure 1*; time spent in the center quadrant in the open field test, it doesn't look like a significant difference between wild-type and Lrp4 mutant mice from the data scatter or error bars. Are these really*
*significantly different?*

We mistakenly illustrated the standard deviation rather than the standard error from the mean values in Figure 1. We have corrected this error and indicated the p value (**p<0.01) for the difference between the mean values in Figure 1.

*3) In*
Figure 5
*G, and H; wild-type mice have approximately -20 pA pre TBS amplitude, and ko approximately -30 pA. Although these can't be compared directly, it makes one wonder if there is a difference in evoked synaptic transmission/ input-output curves for fiber volley. This is important to test because the decrease in mEPSC frequency and spine density suggest a decrease in synapse number in the Lrp4m mutants*.

There is considerable variability in the baseline EPSC amplitudes even in slices from wild-type animals. There was not a significant difference in the baseline EPSC amplitudes between wild-type and mutant mice (wild-type, 52.6 ± 9.7 pA, n=21; *Lrp4*^*-/-*^*; Lrp4*^*m*^*,* 44.9 ± 4.9 pA, n=27, p=0.44). There was not a significant correlation between the magnitude of LTP and initial synaptic strength (wild-type, R^2^=0.05, n=21, p=0.35; *Lrp4*^*-/-*^*; Lrp4*
^*m*^, R^2^=0.09, n=21, p=0.13). This information has been added to the legend for Figure 5.

In the new Figure 6, we now show EPSC strengths from wild-type and mutant mice as a function of extracellular stimulation intensity.

*4) Also in*
Figure 5*, panel G and H, post TBS, the red bar in wt is approximately -55 (compared to -20 pre TBS), and in ko is -40 (compared to -30 pre TBS), but the % baseline values in L don't seem to reflect these values. Can the authors describe how they normalized panels G and H to generate panel L? Likewise, in panel J there's a change from -70 to -160 (black bar compared to red bar) in wt and in panel K from -70 to -120 in ko, which may be significantly different. But in the comparison in panel M, both overlap at*
*approximately 160% potentiation?*

For Figure 5N, we computed the magnitude of LTP as the average synaptic strength 10-20 minutes after pairing, normalized by the average synaptic strength before pairing (Feldman, Neuron 2000; Froemke et al., Nature 2005; Gambino et al., Nature 2014). We have modified the Materials and methods section to include this description. It is apparent from Figure 5N that there is some cell-to-cell variability in the amount of LTP induced for individual recordings (e.g., some EPSCs are more than doubled in amplitude, while other EPSCs are only modestly affected), as is typical for these types of experiments.

As in our response to question #3 above, although these particular examples expressed 228% (wild-type cell in Figure 5J) and 160% LTP (KO cell in Figure 5), on average both groups had comparable magnitudes of LTP when theta-burst stimulation was paired with postsynaptic depolarization (wild-type, 166.3 ± 19.3 %; KO, 157.5 ± 22.4%).

*5) Do the authors propose two different effects of Lrp4? In synapse formation and plasticity? Or do they think the mEPSC frequency decrease and spine number decrease are related to the LTP deficit? To test synapse formation in more detail they could quantify number of synapses (i.e. pre and post-synaptic sites marked with synaptophysin or PSD-95; in slices or in hippocampal cultures) in wild-type and Lrp4m. This would also distinguish between the mEPSC frequency being caused by a change in synapse number or release probability*.

We have proposed that the change in mEPSC frequency and failure to induce LTP may be due, at least in part, to the decrease in number of inputs to the apical dendrites of CA1 neurons. We visualized dendritic spines on CA1 neurons using *Thy-1 GFP* mice in which neurons are sparsely labeled. Although double-staining for synaptophysin and PSD-95 would be an alternative approach for measuring synapse number, we believe that the reduction in the number of spines on apical dendrites indicates that the number of synapses in this region is reduced.

*6) A possible cause of the LTP deficit in Lrp4m kos, on the other hand, could be a decrease in the strength of excitatory synapses in knockouts. In the Discussion the authors state that inhibition may have been strengthened in Lrp4 kos. What is the basis for this statement? There was a significant decrease in GABA in the kos (which is not commented on). It would be useful to measure inhibition, and possible changes in GABA at synapses in addition to test for possible changes in excitation/ inhibition ratio, which could explain the LTP phenotype. In addition, it would be helpful to know if Lrp4 is expressed in all neurons or in particular subsets of neurons in the hippocampus where it might change feedback or feed-forward inhibition, for example, or if it acts more generally at all synapses*.

We had not carried out experiments or supplied data to demonstrate that inhibition was altered in *Lrp4*^*-/-*^*; Lrp4*^*m*^ mice. In the Discussion we raised this possibility because enhanced inhibition and depolarization-induced suppression of inhibition (DSI) could have explained a lack of LTP and restoration of LTP by depolarizing CA1 neurons (Pitler and Alger, 1994; Wilson et al., 2001).

During the past month, we carried out experiments to directly test this idea, and the new data are presented in a new Figure 6 and discussed in the revised text.

“However, we also considered the possibility that in the absence of Lrp4, inhibition on CA1 neurons may have been strengthened, and depolarization-induced suppression of inhibition (DSI) may have removed the enhanced inhibition, unmasking LTP (Pitler and Alger, 1994; Wilson et al., 2001). We therefore directly measured inhibition to determine whether it was strengthened in *Lrp4*^*-/-*^*; Lrp4*^*m*^ mice (Figure 6). Contrary to this notion, we found that inhibition was modestly reduced at low stimulation intensities and unchanged at higher stimulation intensities in *Lrp4*^*-/-*^*; Lrp4*^*m*^ mice (Figure 6). Because excitation was reduced while inhibition remained largely unaffected (Figures 5 and 6), the E/I ratio at CA1 synapses was diminished (Figure 6). Together, these data are inconsistent with the idea that enhanced inhibition is responsible for a failure to elicit LTP and instead favor the idea that a failure of presynaptic input to adequately depolarize CA1 neurons underlies the LTP deficit.”

In situ hybridization studies of *lrp4* in the adult rodent brain ([41]; Allen Brain Atlas) show that *lrp4* is expressed in the granule cell layer of the dentate gyrus and in each of the CA fields of the hippocampus.

Reviewer 2:

*This is a very well designed study documenting importance of LRP4 in central nervous system and especially for long-term plasticity, learning and memory. The findings are based on well-designed behavioral function tests, direct recordings from hippocampal neurons, and examination of dendritic ultrastructure. The altered synaptic plasticity is attributed to decreased dendritic spine density and reduced mEPSC frequency. I only have two comments, one of which is minor*.

*1) Avoid abbreviations in the*
*Abstract (e.g., spell out LTP)*

We have replaced LTP with long-term potentiation.

*2) LRP4-seropositive myasthenic patients are unlikely to develop neurological deficits in central nervous system because autoantibodies do not cross the blood-brain barrier unless it is inflamed or otherwise damaged. This interpretation should be omitted in the Abstract and the Discussion*.

For the reasons outlined below, we prefer to keep this interpretation in our manuscript.

In several autoimmune diseases that ultimately attack targets in the CNS, there is no evidence for initial attack on the brain, but on a peripheral target, such as a tumor. Paraneoplastic diseases are clear examples of antibodies crossing the blood-brain barrier (BBB), although the disease and antibodies are elicited in the periphery (Leypoldt et al. 2014 Autoimmune encephalopathies. Ann NY Acad Sci doi: 10.1111/nyas.12553, Epub ahead of print). For example, Dalmau’s studies have shown that autoimmune encephalitis can arise from production of antibodies to a variety of neuronal cell surface proteins, including the NMDA receptor, LG1 and Caspr2 (Armangue et al. 2014 Autoimmune encephalitis as differential diagnosis of infectious encephalitis. Curr Opin Neurol 27:361; Rosenfeld et al. 2012 Paraneoplastic syndromes and autoimmune encephalitis: Five new things. Neurol Clin Pract 2:215 Graus and Dalmau 2012 Paraneoplastic neurological syndromes. Curr Opin Neurol 25:795). Antibodies to the NMDA receptor are commonly provoked by an ovarian teratoma, and antibodies to LG1 and Caspr2 are often triggered by a thymoma (see reviews cited above).

Antibodies to these neuronal cell surface proteins are present in the cerebral spinal fluid (CSF), suggesting that the antibodies are responsible, at least in part, for the encephalitis. Similarly, patients with Lambert-Eaton myasthenia and antibodies to P/Q calcium channels develop cerebellar ataxia, indicating that the antibodies cross the BBB. Although IgG levels are normally 100 to 200-fold lower in the CSF than in the blood, it seems likely that the low level of antibody that normally crosses the BBB and enters the CSF can initiate a response, possibly triggering a further breakdown in the BBB (Graus et al. J. 2010. Antibodies and neuronal autoimmune disorders of the CNS. Neurol. 257:509). For these reasons, we believe that it is possible that antibodies to Lrp4 in patients with Lrp4 myasthenia gravis may cause neurological deficits.